# From larva to adult: *In vitro* rearing protocol for honey bee (*Apis mellifera*) drones

**Marina Carla Bezerra da Silva**[1]*, **Madison Gail Kindopp**[1], **Midhun Sebastian Jose**[1], **Oleksii Obshta**[1], **Thanuri Lakna Kumari Edirithilake**[1], **Emilio Enrique Tellarini Prieto**[1], **Muhammad Fahim Raza**[1], **Marcelo Polizel Camilli**[1], **Jenna Thebeau**[1], **Fatima Masood**[2], **Ivanna Kozii**[3], **Igor Moshynskyy**[1], **Elemir Simko**[1], **Sarah C. Wood**[1]

1 Department of Veterinary Pathology, Western College of Veterinary Medicine, University of Saskatchewan, Saskatoon, Saskatchewan, Canada, 2 Department of Veterinary Microbiology, Western College of Veterinary Medicine, University of Saskatchewan, Saskatoon, Saskatchewan, Canada, 3 Prairie Diagnostic Services Inc, Western College of Veterinary Medicine, University of Saskatchewan, Saskatoon, Saskatchewan, Canada

* marina.silva@usask.ca

**Data Availability Statement:** All relevant data are within the manuscript and its Supporting Information files.

## Abstract

Development of a successful *in vitro* rearing protocol has been essential for pesticide safety assessment of immature honey bee workers under laboratory conditions. In contrast, pesticide safety testing of honey bee drones is limited, in part due to the lack of successful laboratory rearing protocols for this reproductive caste. Considering that healthy drones are essential for successful mating and reproduction of the honey bee queen, a standardized *in vitro* rearing protocol for honey bee drones is necessary to support reproductive safety studies, as well as to gain a deeper understanding of honey bee drone development. Using the established *in vitro* rearing protocol for honey bee workers, we modified the days of grafting and pupal transfer, as well as the diet volume, pupation plate orientation, and absorbent tissue in the pupal wells to successfully rear honey bee drones *in vitro*. *In vitro*-reared drones were evaluated for gross wing abnormalities, body weight, testes weight, and abdominal area, and compared with age-matched drones reared in field colonies. We found that honey bee drones reared in a vertically oriented pupation plate containing WypAll® absorbent tissue in each well had a mean survival to adulthood of 74 ± 3.5% (SEM) until adulthood. In contrast, drones reared in a horizontally oriented pupation plate containing Kimwipe® absorbent tissue in each well had significantly lower survival (5.5 ± 2.3%) and demonstrated gross wing abnormalities. All *in vitro*-reared drones had significantly lower body weight, testes weight and abdominal area relative to colony-reared control drones. Accordingly, we successfully developed an *in vitro* rearing protocol for honey bee drones which has the potential to improve future reproductive safety assessment of pesticides for honey bees.

## Introduction

*In vitro* rearing of honey bee workers from larva to adult has been an essential methodology for the evaluation of teratogenic effects of xenobiotic exposure on worker development. For

**Funding:** Funding acquired by SW and ES Bayer Research & Development Services, LLC https://www.cropscience.bayer.ca/en/ The funders had no role in study design, data collection and analysis, decision to publish, or preparation of the manuscript.

**Competing interests:** The authors have declared that no competing interests exist

instance, exposure of honey bee workers to pesticides during *in vitro* development [1–3] has enabled the identification of potential risks and later, guided regulators in recommending best management practices for pesticide application to minimize negative impacts on non-target pollinators [4].

Standardized pesticide risk assessment for honey bees is currently limited to three tiers [5] including:

- Laboratory determination of the $LD_{50}/LC_{50}$ for larvae and adult worker honey bees, and estimation of individual exposure in the environment;

- Semi-field assessment of colonies foraging on a treated crop within an enclosure, including measurement of adult bee mortality, foraging activity, brood production and pesticide residues in colony food stores and

- Field assessment of free-flying colonies foraging on treated crops, including evaluation of colony strength, behavior, disease or pest levels, pesticides residues, and food stores in colonies.

Risk assessment proceeds in a stepwise when acceptable risk cannot be determined in the previous tier. Importantly, tier-1 laboratory pesticide risk assessment procedures have been critically enhanced by the development of *in vitro* rearing protocols for honey bee workers [6]. Specifically, *in vitro* rearing protocols have enabled acute and chronic pesticide exposure assays for worker larvae [5], which have improved our overall understanding of pesticide ecotoxicology to multiple life stages of honey bees [3].

Unfortunately, to date, there is no established protocol for *in vitro* rearing of the reproductive castes of honey bees, including drones and queens [7–9]. Honey bee drones are the only male bees in the hive and, as such, are essential for the fertilization of the queen and the preservation of genetic diversity in the honey bee population [10]. Although honey bee drones are not responsible for foraging in the environment, drones are nevertheless exposed to agrochemicals that accumulate within hive matrices, such as in wax and stored pollen [11].

Concerningly, chronic exposure of drones to pesticides has negative effect on drone reproductive fitness. For instance, developing honey bee drones chronically exposed to a combination of neonicotinoids (thiamethoxam and clothianidin) were found to have significantly reduced sperm count by 39% [12]. Moreover, chronically exposure during development to the combination of miticides fluvalinate, coumaphos and amitraz, and the agrochemicals chlorothalonil and chlorpyrifos in wax was shown to significantly decrease drone sperm viability [13]. Most studies of the effects of xenobiotic exposure on adult honey bee drones [12, 13] have used in-hive exposure models which are notoriously difficult to control and achieve consistent xenobiotic exposure. Accordingly, a method for *in vitro* rearing of honey bee drones is urgently needed to facilitate standardized evaluation of xenobiotics for reproductive effects in honey bee drones.

Despite several attempts to develop an *in vitro* rearing protocol for honey bee drones [7, 8, 14–16], to date, the maximum reported drone survival to eclosion was 28% [7]. In contrast, validating *in vitro* rearing protocols for honey bee workers requires ≥ 70% survival to adulthood in the control group [5]. Various modifications to diet and environmental conditions have been investigated to improve drone survival *in vitro*. For example, increasing the amount of royal jelly in the larval diet and decreasing the incubation temperature and relative humidity have been investigated to improve drone rearing success [8, 17].

Previously developed *in vitro* rearing methods for honey bee drones have yielded valuable insights; for example, a study by [7] demonstrated the importance of including fructose in the drone diet. However, these protocols have limitations, such as the lack of detailed information

on diet volume provided, which reduces their replicability. Additionally, despite various modifications in rearing methods, none of the previous studies have accounted for key aspects of the natural honey bee environment, such as the orientation of comb cells and cleaning behaviors, which may be critical to drone development.To address the current lack of successful protocols for *in vitro* rearing of honey bee drones, the goals of this study were to (1) identify optimal feeding and environmental conditions to successfully rear honey bee drones *in vitro* from larva to adult with $\geq$ 70% survival, comparable to *in vitro* rearing of honey bee workers, and (2) to describe the characteristics of *in vitro*-reared drones, including wing morphology, body weight, testes weight, and abdominal area, in comparison to drones reared in field colonies.

## Materials and methods

### 1. Ethics statement

This study did not involve the use of endangered or protected species.

### 2. Study design

We adapted the *in vitro* rearing protocol for honey bee workers developed by Schmehl *et al*. [6] to rear honey bee drones *in vitro* according to the following modifications (Fig 1):

- Increasing the age of larval grafting from four days after queen caging (first instar larva) to five days after queen caging (second instar larva)

- Modifying the larval feeding schedule to increase the total diet volume by 2.3-fold

- Delaying the experimental day of prepupal transfer by one day (from D6 to day D7)

- Altering the pupal plate orientation from horizontal to vertical

- Changing the absorbent tissue in the pupal wells from Kimwipe® to WypAll® Economizer L30 1⁄4 Fold Wipers (Uline.ca, Edmonton, Alberta, Canada)

Drone survival was recorded daily. Gross wing abnormalities, prepupal and adult body weight, testes weight, and abdominal area of *in vitro*-reared drones were compared to age-matched drones collected from field colonies.

### 3. Honey bee drone larvae production

We generated age-synchronized frames of honey bee drone larvae for *in vitro* rearing from June to mid-August, 2023 using six, healthy honey bee colonies from the research apiary at the University of Saskatchewan in Saskatoon, Saskatchewan, Canada. Every 24 hours an empty, wax-drawn, drone brood frame was inserted into a cage within each colony where the queen was confined. After 24 hours, frames with eggs were removed from the cage and incubated in the adjacent brood chamber of the same colony for five days, after which the frames of second instar larvae were transferred to the lab in an insulated container containing hot water bottles to maintain the temperature at approximately 33°C - 35°C.The frames were kept no more than 1 hour in portable incubator prior to grafting. To avoid contamination, the portable incubator was cleaned daily before use with Prevail™ disinfectant (Horizon Livestock & Poultry Supply, Steinbach, Manitoba, Canada) in a 1:40 (v:v) with a one minute contact time.

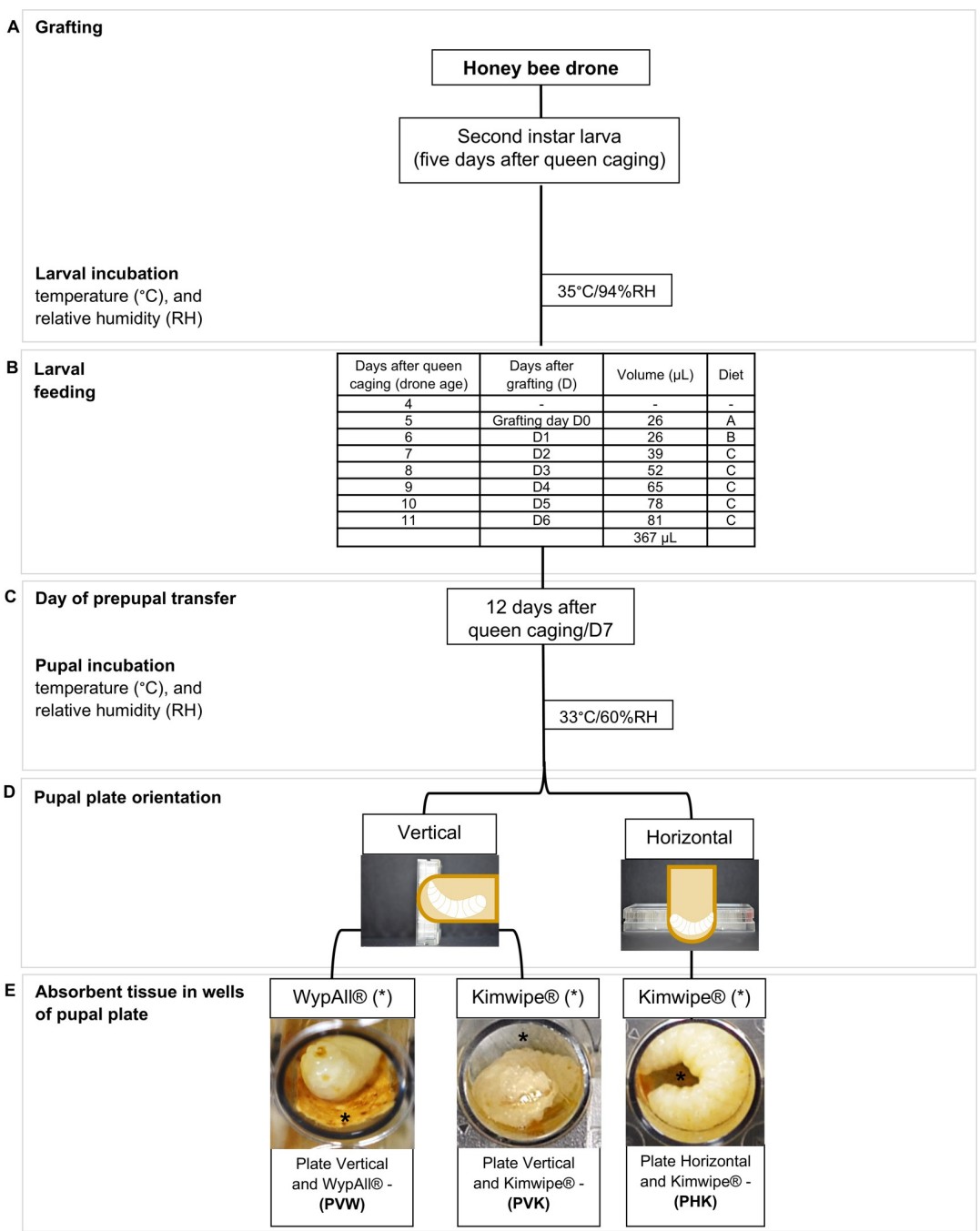

**Fig 1. *In vitro* rearing protocol for honey bee (*Apis mellifera*) drones.** (A) grafting and larval incubation, (B) larval feeding, (C) day of prepupal transfer and pupal incubation, (D) pupal plate orientation, and (E) absorbent tissue in wells of pupal plate.

## 4. Equipment and supplies

We used the same materials and supplies outlined in the protocol for *in vitro* rearing of honey bee workers [6] with the following modifications:

**4.1. Grafting.** A metal German grafting tool (Dancing Bee Equipment, Winnipeg, Manitoba, Canada) was used to transfer larvae from the brood frame to 48-well sterile tissue culture plate (STCP) wells and to transfer prepupae to new STCP wells for pupation.

**4.2. Larval feeding.** To accommodate the larger volume of larval diet used in this study, we used step pipettes with a range of 0.05 mL to 50 mL (Corning® Step-R™ Repeating Pipettor) and 0.5 mL to 50 mL (Thermo Scientific™ Finnpipette Stepper Pipette), with the manufacture recommended tips for that volume range. The STCP was placed at a 45-degree angle on an electric heating pad set to 35°C during larval feeding.

**4.3. Pupation.** During pupation, the wells of the STCP were lined with either Kimwipe®, as described for workers [6], or WypAll® Economizer L30 1⁄4 Fold Wipers (Uline.ca, Edmonton, Alberta, Canada). Parafilm strips were used to secure the STCP lids when plates were oriented vertically.

**4.4. Decontamination.** We applied Prevail™ disinfectant in a 1:40 (v:v) dilution with a 1 minute contact time, followed by 30 minute exposure to UV light and application of 70% alcohol to the equipment and surfaces in the biological safety cabinet prior to grafting, feeding, pupal transfer, and survival assessment.

## 5. Methods

**5.1. Grafting.** The frames with larvae brought from the field to the lab were kept for no more than 1 hour in the portable incubator previous grafting, as described previously. Using a biological safety cabinet running at half speed (vent speed) to minimize drying of larvae, second instar (five days post-oviposition) drone larvae were individually transferred (grafted) from the drone brood frame to 48-well STCPs. Each well of the STCP contained a plastic cell cup [6] filled with 26 μL of control diet 'A' (see 4.2) [6] pre-warmed to 35°C. During grafting, the STCPs were kept on an electric heating pad at 35°C. After grafting, the STCPs were placed in an incubator set at 35°C (34.7 ± 0.5°C) within a desiccator containing salt solutions of $K_2SO_4$ to maintain a relative humidity (RH) of 94% (93 ± 10%) [6]. Temperature and RH were in the larval desiccator monitored hourly by an Onset HOBO MX1101 Wireless Temperature/ Humidity sensor (ITM instruments Inc., Sainte-Anne-de-Bellevue, Canada).

**5.2. Larval feeding.** Larval diets were prepared using the recipes by Schmehl et al. [6] established for honey bee worker larvae. In summary, we prepared three diets, labeled 'A', 'B', and 'C' using sterile royal jelly (Stakich Inc., Troy, MI, United States), glucose, fructose, yeast extract, and sterile distilled water. The diets were stored at −20°C until use.

The feeding schedule and volumes (Fig 1B) were based on the protocol for *in vitro* rearing of honey bee workers [6] with the following adjustments:

1. Larvae were fed daily until prepupal transfer, including experimental day 1 (D1).

2. Larval diet volume was increased by 30% from the recommended worker larval diet volume [6] from D0 to D4. Since worker larvae are not fed on D1, the diet volume fed to drones on D1 was determined by increasing the volume fed to workers on D2 by 30%. Correspondingly, on D2, D3, and D4, drones were fed 30% more diet compared to workers fed on D3, D4, and D5, respectively (Fig 1B). On D5, drones were fed 30% more than the calculated volume that worker larvae would be fed on D6 (60 μL), presuming worker larval feeding continued past D5 (worker larval diet volume increases by 10 μL increments per day [6]).

3. Drones were fed for one additional day (D6) to account for their biologically longer larval development (eight days) relative to honey bee workers (seven days [18]). The volume of diet to feed on D6 was determined empirically. We found that larvae could finish consuming 81 μL of their food before being transferred to a new STCP for pupation on D7.

4. The total diet volume fed was 367 μL for drones, representing a 2.3-fold increase in the total diet fed relative to workers (160 μL). This increase in diet approximates the reported difference in weight of drones (400 mg) vs. workers (162 mg) at the end of larval development (2.5-fold difference) [8, 19].

**5.3. Pupation.** Drones were transferred to the pupal plate one experimental day later (D7, 12 days post-oviposition, first day of pre-pupal development) relative to workers (D6, 10 days post-oviposition, last day of larval development) to account for the biologically longer larval development of drones. Only those prepupae that had completely consumed their diet were relocated to the pupation STCP, while the remaining prepupae were retained in the larval STCP until all diet was consumed.

Drones were reared in the pupation plates in two orientations: 1) STCP was oriented vertically (90° to the horizontal plane) or 2) STCP was oriented in the horizontal plane (Fig 1D). The vertical orientation was included to simulate the natural orientation of developing drones in a wax cell in a hive, where the brood cells are normally oriented approximately 13˚ from the vertical plane [18]. To reduce fecal soiling of the STCP lids in the vertically oriented plates, lids were cleaned every 24 hours with Prevail™ disinfectant with a one-minute contact time.

Similar to the rearing protocol for honey bee workers [6], for the horizontally oriented STCPs, the bottom of each well contained a 2.0 × 1.0 cm piece of cut out of Kimwipe® to reduce fecal contamination of the developing drone pupae (denoted hereafter as 'PHK' for Plate Horizontal with Kimwipe®). For STCPs oriented vertically, two types of absorbent tissue lining the bottom and lateral sides of the well were tested (Fig 1E): (1) Kimwipe® liner measuring 2.0 × 1.0 cm (denoted hereafter as 'PVK' for Plate Vertical with Kimwipe®) and (2) WypAll wiper® liner measuring 1 x 2.8 cm (denoted hereafter as 'PVW' for Plate Vertical with Wypall®).

To mimic natural atmospheric conditions in the hive where drones are typically found at the edges of the brood nest, during pupation, drones were maintained at a lower temperature (33˚C) and RH (60%) relative to honey bee workers (35˚C and 75% RH) (Fig 1C). In contrast to worker pupal STCPs [6], drone pupal STCPs were not kept in a desiccator, but instead maintained in a temperature and humidity-controlled insect-rearing chamber (Caron, 7340–25 model, VWR international, Edmonton, AL) which was cleaned weekly with Prevail™ disinfectant.

## 6. Outcome measures

**6.1. Survival.** Every 24 hours, a stereomicroscope (Olympus system SZ61/SZ51, Leica Microsystems Inc, Ontario, CA) was used to evaluate larval survival, while the prepupa and pupal viability were assessed by eye. Dead larvae, prepupae, and pupae were identified by a flaccid, shrunken, dull appearance with multifocal brown-black coloration and lack of movement in the larval or late pupal stages. Adult drones were considered successfully emerged if they moved out of their well in the STCP.

**6.2. Wing abnormalities, body weight, testes weight, and abdominal area.** Wing abnormalities, such as absence (aplasia) or reduced size (hypoplasia), were visually evaluated in all adult drones that successfully emerged in the PVK and PVW groups (166/240 survived to adult eclosion). Furthermore, wing morphology was assessed in adult drones that successfully emerged in the PHK group (5/92 survived to adult eclosion).

For the PVW group, body weight was measured using an analytical balance at the time of transfer to pupal STCP on D7 (n = 21) and at adult emergence (n = 30). Before measuring body weight, adult drones were euthanized by immersion in 10% formalin for 48 hours and

blotted dry. Additionally, the abdomen length and width were measured at the widest point using a digital caliper and the abdominal area (mm$^2$) was calculated by multiplying the length and width for each drone. After weighing, the adult drones were dissected to remove the paired testes. Briefly, the left lateral and dorsal abdominal cuticle was removed to allow visualization and removal of the testes, and the combined weight of both testes from each drone was determined using an analytical balance.

As a control for comparison to the PVW group, age-synchronized drones from three brood frames (representing three genetic lines) were reared in three field colonies. At day 12 post-oviposition (DP12; equivalent to D7 *in vitro*) twenty prepupae (6–7 per genetic line) were collected and in colony and *in vitro* prepupae fresh body weight (non-formalin fixed) was determined using an analytical balance. At DP22, the three drone frames were transferred to a laboratory incubator at a temperature of 33˚C and 60% RH and 10 emerging adult drones were collected per frame (n = 30) on DP24. As described above for the *in vitro*-reared PVW group, body weight, abdominal area, and testes weight were determined after fixation in 10% formalin for 48 hours.

## 7. Statistical analysis

Stata version 17 (StataCorp LLC, College Station, TX, United States) software was used for all statistical analyses. All tests were 2-tailed, and our α was set at 0.05. Normality was assessed using the Shapiro-Wilk test and equality of variance was assessed using Levene's test. Fisher's exact test was used to compare percent survival to adult emergence among the PVW, PVK, and PHK groups and log-rank test was used to compare the survival over time among groupsPrepupa body weight (0.92<W<0.95, P = 0.10, P = 0.52), adult body weight (0.94<W<0.95, P = 0.11, P = 0.24), abdominal area (0.95<W<0.97, P = 0.19, P = 0.63) and testes weight (0.95<W<0.97, P = 0.20, P = 0.57) values were normally distributed assessed by Shapiro-Wilk test. Mean prepupa body weight and mean adult testes weight were compared between the PVW group reared *in vitro* and control drones reared in field colonies using Welch's approximate t-test, since these analyses did not have equal variance (Levene's test for prepupa body weight: $F_{19,20}$ = 13.07, P< 0.001; Levene's test for adult testes weight: $F_{29,29}$ = 3.37, P = 0.001). Mean adult body weight and mean adult abdominal area were compared between the PVW group reared *in vitro* and control drones reared in field colonies using an independent t-test. The two groups met the assumption of equal variance (Levene's test for adult body weight: $F_{29,29}$ = 1.70, P = 0.156; Levene's test for abdominal area: $F_{29,29}$ = 1.59, P = 0.215).

## Results

We successfully developed a standardized protocol to rear honey bee drones *in vitro* from second instar larvae (five days after queen caging; Day 5) to adult eclosion with a mean survival of 74% ± 3.5% (SEM) for drones reared in the vertically oriented pupation plates containing a WypAll$^®$ absorbent liner (Fig 2) (S1 File). We found that grafting drone larvae at day 5 after queen caging improved adult drone survival to adult eclosion relative to the grafting of larvae at day 4 (data in S1 File), which is recommended for *in vitro* rearing of honey bee workers [6]. Even grafting drones on day four and orienting the plates vertically during pupation, as was done on day five, did not improve survival (S1 Fig).

We found that modifications to pupation plate orientation and absorbent tissue in the pupal wells significantly improved survival to adulthood. Drones reared in pupation plates oriented vertically with Kimwipe$^®$ absorbent tissue (PVK) had significantly increased survival to eclosion, by 167% (Fisher's Exact Test; P<0.001) (Fig 2), relative to drones reared in pupation plates oriented horizontally with Kimwipe$^®$ absorbent tissue (PHK). Moreover, for drones

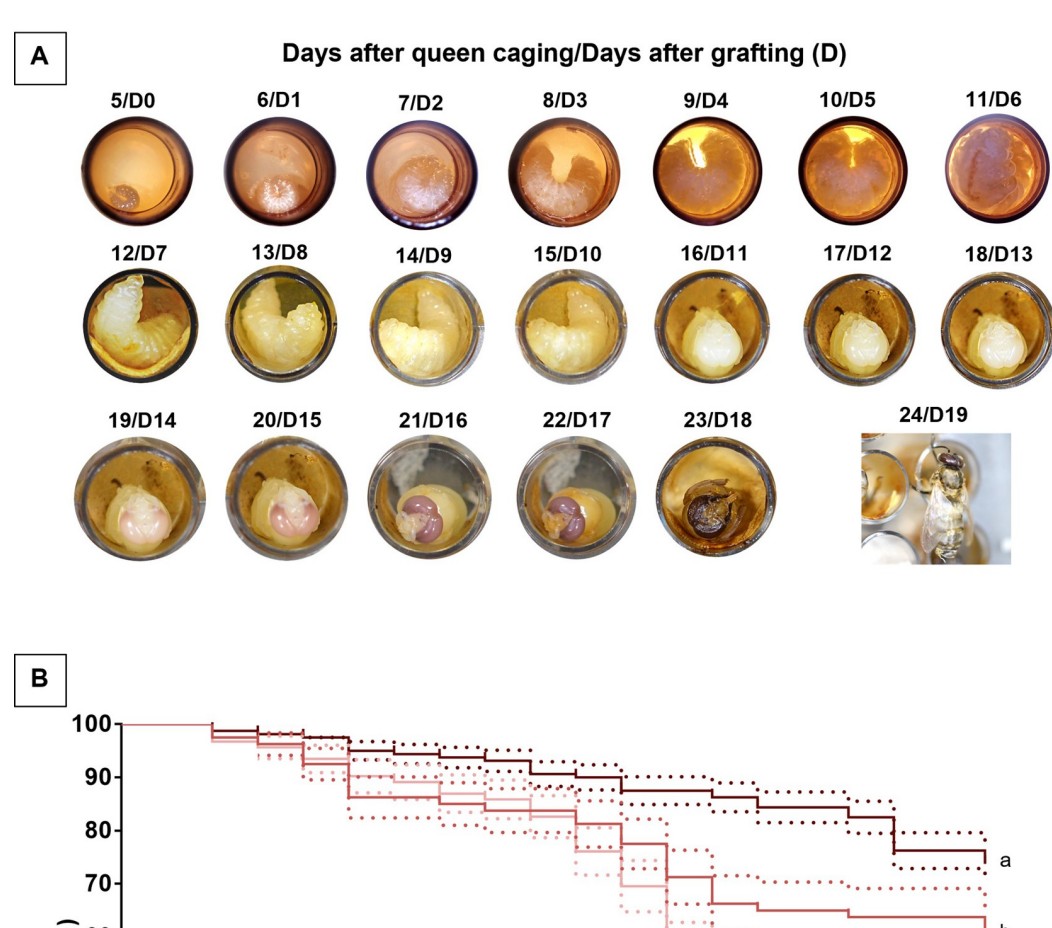

**Fig 2. Daily development photo and survival of *in vitro* reared honey bee drones.** A) Daily images of honey bee drones reared *in vitro* in the vertical plate orientation with WypAll® (PVW) and vertical plate orientation with Kimwipe® (PVK). B) Percent survival (± SEM in dashed lines) over time in days (post-oviposition) of honey bee drones reared *in vitro* in either the horizontal plate orientation with Kimwipe® absorbent tissue (PHK; n = 92) or the vertical plate orientation with WypAll® (PVW; n = 160) or Kimwipe® (PVK; n = 80). Different letters signify statistical differences at α = 0.05.

reared in the vertically oriented pupation plates, the use of a WypAll® absorbent liner in the pupation wells (PVW) significantly increased survival to eclosion by 21% relative to drones reared in wells containing Kimwipe® absorbent liner (Fisher's Exact Test; P = 0.037) (Fig 2).

All drones reared in pupal plates oriented horizontally (PHK group) and surviving to adult eclosion (5/92 survived) had abnormalities in wing development, including hypoplasia (Fig 3B) or aplasia (Fig 3C). In contrast, wing abnormalities were not observed in the 166 adult drones that survived to eclosion in the PVW (118/160 survived) and PVK groups (48/80 survived) (Fig 3A).

Relative to age-matched drones reared in field colonies, we found that PVW drones reared *in vitro* had significantly reduced body weight as prepupae and newly emerged adults (Fig 4A), as well as significantly reduced adult abdominal area and testes weight (Fig 4B) (data in S2 and S3 Files). Prepupae reared *in vitro* had significantly lower body weight, by 12% or 42 mg on average, relative to colony-reared controls (Welch's approximate t-test, $t_{39}$ = 2.952, P = 0.005), while adult drones reared *in vitro* had a significantly lower body weight, by 23% or 77 mg on average, *relative* to colony-reared controls (independent t-test, $t_{58}$ = 11.2, P<0.001). Furthermore, the abdominal areas of *in vitro*-reared adult drones were significantly smaller, by 8% or 3 $mm^2$ on average, relative to colony-reared controls (independent t-test, $t_{58}$ = 2.91, P = 0.005). Finally, the adult testes weight of *in vitro*-reared adult drones was significantly lower, by 52% or 15 mg on average, compared with colony-reared controls (Welch's approximate t-test, $t_{58}$ = 13.65, P<0.001).

## Discussion

We successfully reared drones from larva to adult with mean survival of 74% ± 3.5% (SEM), by modifying the protocol for *in vitro* rearing of honey bee workers, including delaying the day of grafting and transfer to the pupation plate, increasing the larval diet volume, re-orienting the pupation plate to the vertical plane, and lining the pupal wells with Wypall® absorbent tissue.

We found that grafting drone larvae at five days post-oviposition (DP5) improved adult drone survival to adult eclosion (Figs 2B and S1) because the larvae grafted at DP5 were larger compared with DP4, which made it easier to see them and achieve successful grafting. Another drone rearing protocol [7] reported grafting at DP4, and demonstrated larval survival up to 90%; however, the maximum survival to adult eclosion reported was 28% In contrast, the higher rates of survival to adult eclosion in our study, up to 74%, may in part have resulted from delaying the day of grafting to DP5.

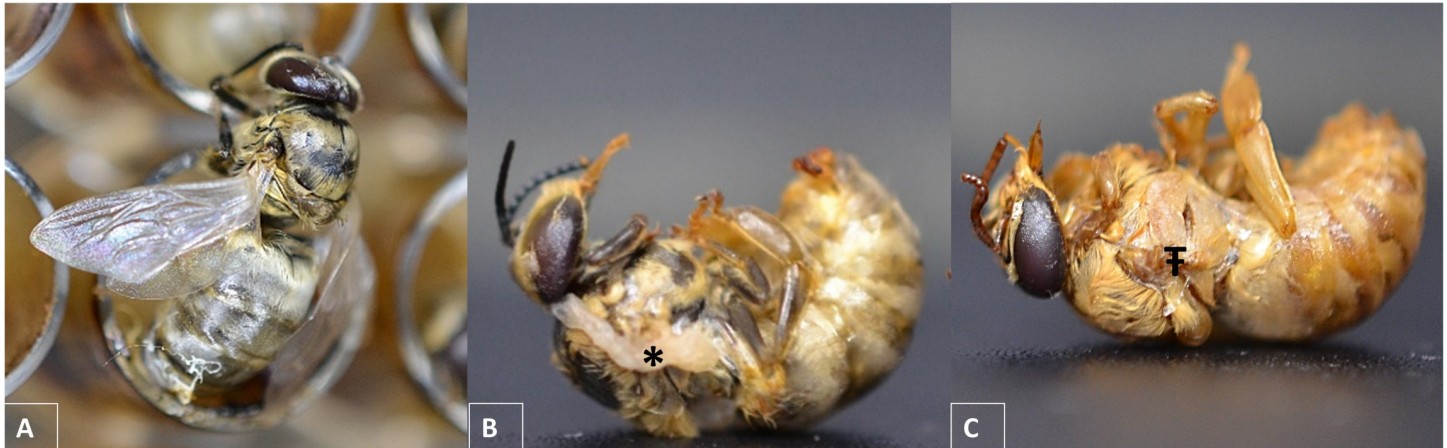

**Fig 3. Gross wing morphology of representative honey bee drones reared *in vitro*.** A) Adult drone emerged from a pupal plate oriented vertically and demonstrating normal wing morphology. B) Adult drone emerged from a pupal plate oriented horizontally and demonstrating wing hypoplasia (*) or C) wing aplasia ().

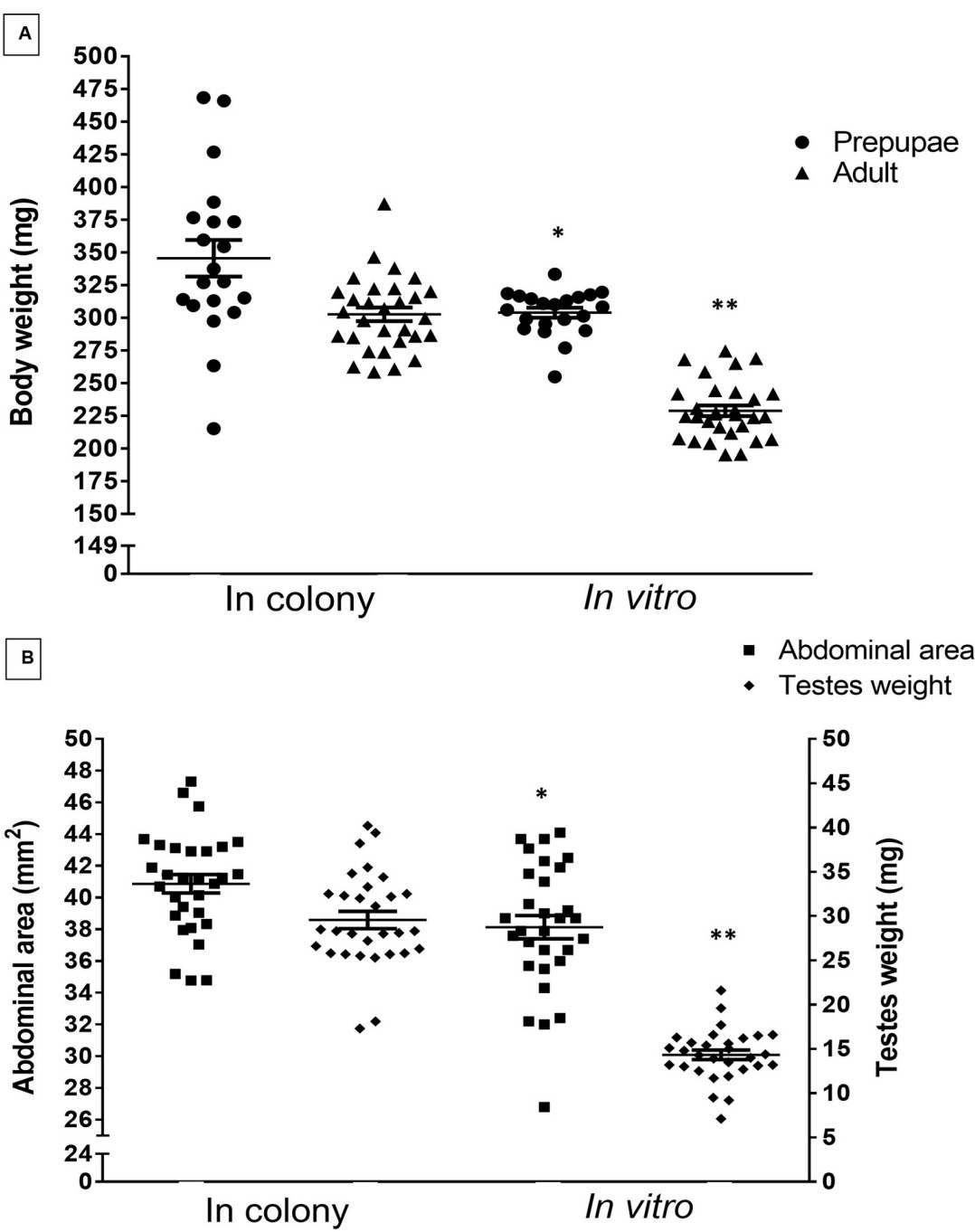

**Fig 4. Body weight, adult testes weight, and abdominal area measurements in honey bee drones reared *in vitro*.** A) Mean ± SEM body weight (mg) and B) mean ± SEM abdominal area (mm²) and testes weight (mg) of age-matched honey bee drones reared either within a colony or *in vitro*. *, ** denotes statistical difference with P<0.01 and P<0.001, respectively, between corresponding life stages reared in a colony (n = 20 prepupae; n = 30 adults) or *in vitro* (n = 21 prepupae; n = 30 adults).

Our larval feeding protocol (Fig 1B) diverges from other published drone-rearing methods, which describe ad libitum feeding [8, 7, 15]. In contrast to the worker larval diet [6] used in this study, other drone-rearing protocols describe larval diets composed of only royal jelly,

resulting in a 47.6% survival after 5 days of feeding [8]. Other reported diet variations include royal jelly supplemented with honey or sucrose, resulting in reported survival to eclosion of 28% and 8%, respectively [7], as well as royal jelly with glucose, fructose, and distilled water, resulting in 90% survival until the prepupa stage [15].

Re-orientation of the pupal plate from the horizontal to the vertical plane was a critical intervention in this study associated with enhanced adult drone eclosion and absence of wing abnormalities. In contrast, plate orientation does not appear to affect survival of adult workers *in vitro*. Kim et al. [20] found that workers reared in larval and pupal plates in the vertical plane had increased survival by 1.4% compared to plates oriented in the horizontal plane. However, similar to drones reared horizontally in our study, workers reared in the horizontal plane had abnormal wing morphology and reduced wing lengths, in addition to smaller body sizes and smaller abdomens [20]. Moreover, a previous study which reared drones in the horizontal plane found that 30% of adult drones emerged *in vitro* had incomplete ecdysis and 60% had curled wings [7]. Although wing abnormalities were correlated with horizontal pupal plate orientation in our study and others, we cannot exclude the possibility of infection of experimental drones with Deformed Wing Virus, which is known to manifest clinically with wing deformities and shortened abdomens [21].

Alteration of the pupal plate orientation in this study may have enabled improved survival to eclosion by providing more physical space for the drone to develop, which may be important considering the larger body size and brood cell size observed in natural drone comb. Future directions for improvement of our drone rearing protocol may include orienting drone pupal plates at 13˚ from horizontal to approximate their natural position within combs in a hive [18] or using larger STCP wells.

We found that use of a WypAll® absorbent liner in the pupal wells significantly improved survival of drones pupating in the vertical plane in contrast to use of a Kimwipe® liner, which is recommended for *in vitro* rearing of workers [6]. Fecal contact is proposed to increase worker pupal mortality [6]; accordingly, the thicker WypAll® tissue may have been more effective at wicking feces away from the pupae. Similarly, Aupinel et al. [22] used cotton dental rolls for fecal absorption during *in vitro* rearing of workers. Another drone rearing protocol [7] attempted to reduce fecal contamination by washing prepupae with lukewarm water and enclosing them in individual paper pupation capsules, although survival to adulthood was only 28%.

Despite our success in rearing drones to adulthood, the *in vitro*-reared drones had significantly reduced body weight, testes weight and abdominal area relative to field-reared controls. By comparison, adult honey bee workers reared *in vitro* were found to have lower body weights and shorter abdomen lengths relative to adult workers emerged in-hive [23]. Other studies of *in vitro* rearing of honey bee drones demonstrated similar adult body weights (236.7 mg [7]) to our study (228.9 mg). Future work will examine whether decreased testes weight of *in vitro*-reared drones is correlated with decreased sperm viability and total sperm count at sexual maturity. As well, future studies will explore a range of drone pupation temperatures, which has been shown previously to influence drone sperm viability and reproductive organ size [17, 24].

In summary, we were able to achieve 74 ± 3.5% *in vitro* survival to adulthood of drone larvae using a modified version of the well-established protocol for rearing honey bee workers *in vitro* [6]. Further refinement of this protocol is necessary to enhance honey bee drone *in vitro* eclosion to greater than 95%, as is often reported for *in vitro* rearing of honey bee workers [6], as well as to increase the body weight and testes weight to match drones reared in-hive.

Nevertheless, the protocol described herein represents a significant step forward for future development of *in vitro* reproductive safety assessment procedures for honey bee drones, as well as improving our understanding of the developmental anatomy and physiology of this caste.

## Supporting information

**S1 Fig. Honey bee drones reared *in vitro* from four days after queen caging.** Percent survival (± SEM in dashed lines) over time in days (after queen caging) in either the horizontal plate orientation with Kimwipe® absorbent tissue (n = 92) or the vertical plate orientation with Kimwipe® (n = 91). Different letters signify statistical differences at α = 0.05.
(TIF)

**S1 File. Survival dataset of honey bee drones reared *in vitro* from four and five days after queen caging.**
(XLSX)

**S2 File. Dataset of adult honey bee drone weights reared *in vitro*.**
(XLSX)

**S3 File. Dataset of adult honey bee drone abdominal area reared *in vitro*.**
(XLSX)

## Acknowledgments

The authors thank Dr. Marco Pietropaoli and Dr. Uros Glavinic for contributions to experimental design and Dr. Angie Magana, Aranza M. Gomez, Debby Peng, Erin E. Baril, Marie Blanchemanche, Mya Desmarais, and Ylona Camus for laboratory assistance.

## Author Contributions

**Conceptualization:** Marina Carla Bezerra da Silva, Madison Gail Kindopp, Midhun Sebastian Jose, Oleksii Obshta, Emilio Enrique Tellarini Prieto, Muhammad Fahim Raza, Marcelo Polizel Camilli, Jenna Thebeau, Ivanna Kozii, Igor Moshynskyy, Elemir Simko, Sarah C. Wood.

**Data curation:** Marina Carla Bezerra da Silva.

**Formal analysis:** Marina Carla Bezerra da Silva, Sarah C. Wood.

**Funding acquisition:** Sarah C. Wood.

**Investigation:** Marina Carla Bezerra da Silva, Madison Gail Kindopp, Midhun Sebastian Jose, Oleksii Obshta, Thanuri Lakna Kumari Edirithilake, Emilio Enrique Tellarini Prieto, Marcelo Polizel Camilli, Jenna Thebeau, Fatima Masood, Igor Moshynskyy, Elemir Simko, Sarah C. Wood.

**Methodology:** Marina Carla Bezerra da Silva, Madison Gail Kindopp, Midhun Sebastian Jose, Oleksii Obshta, Thanuri Lakna Kumari Edirithilake, Emilio Enrique Tellarini Prieto, Muhammad Fahim Raza, Marcelo Polizel Camilli, Jenna Thebeau, Fatima Masood, Ivanna Kozii, Igor Moshynskyy, Elemir Simko, Sarah C. Wood.

**Project administration:** Marina Carla Bezerra da Silva, Sarah C. Wood.

**Resources:** Sarah C. Wood.

**Software:** Marina Carla Bezerra da Silva, Marcelo Polizel Camilli.

**Supervision:** Elemir Simko, Sarah C. Wood.

**Validation:** Marina Carla Bezerra da Silva, Ivanna Kozii, Elemir Simko, Sarah C. Wood.

**Visualization:** Marina Carla Bezerra da Silva, Madison Gail Kindopp, Midhun Sebastian Jose, Oleksii Obshta, Thanuri Lakna Kumari Edirithilake, Emilio Enrique Tellarini Prieto, Muhammad Fahim Raza, Marcelo Polizel Camilli, Jenna Thebeau, Fatima Masood, Ivanna Kozii, Elemir Simko, Sarah C. Wood.

**Writing – original draft:** Marina Carla Bezerra da Silva, Madison Gail Kindopp, Sarah C. Wood.

**Writing – review & editing:** Marina Carla Bezerra da Silva, Madison Gail Kindopp, Midhun Sebastian Jose, Oleksii Obshta, Thanuri Lakna Kumari Edirithilake, Emilio Enrique Tellarini Prieto, Muhammad Fahim Raza, Marcelo Polizel Camilli, Jenna Thebeau, Fatima Masood, Ivanna Kozii, Igor Moshynskyy, Elemir Simko, Sarah C. Wood.

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
