## [Decision Letter · Decision Letter 0]

27 Sep 2024

PONE-D-24-39196Re: From larva to adult: *In vitro* rearing protocol for honey bee (*Apis mellifera*) dronesPLOS ONE

Dear Dr. Wood,

Thank you for submitting your manuscript to PLOS ONE. After careful consideration, we feel that it has merit but does not fully meet PLOS ONE’s publication criteria as it currently stands. Therefore, we invite you to submit a revised version of the manuscript that addresses the points raised during the review process.

We look forward to receiving your revised manuscript.

Kind regards,

Yahya  Al Naggar

Academic Editor

PLOS ONE

Journal Requirements: When submitting your revision, we need you to address these additional requirements. 1. Please ensure that your manuscript meets PLOS ONE's style requirements, including those for file naming. The PLOS ONE style templates can be found at https://journals.plos.org/plosone/s/file?id=wjVg/PLOSOne_formatting_sample_main_body.pdf and https://journals.plos.org/plosone/s/file?id=ba62/PLOSOne_formatting_sample_title_authors_affiliations.pdf 2. We notice that your supplementary figures are uploaded with the file type 'Figure'. Please amend the file type to 'Supporting Information'. Please ensure that each Supporting Information file has a legend listed in the manuscript after the references list. 3. Please include captions for your Supporting Information files at the end of your manuscript, and update any in-text citations to match accordingly. Please see our Supporting Information guidelines for more information: http://journals.plos.org/plosone/s/supporting-information.

Reviewers' comments:

Reviewer's Responses to Questions

**Comments to the Author**

1. Is the manuscript technically sound, and do the data support the conclusions?

Reviewer #1: Yes

Reviewer #2: Yes

2. Has the statistical analysis been performed appropriately and rigorously? 

Reviewer #1: Yes

Reviewer #2: I Don't Know

3. Have the authors made all data underlying the findings in their manuscript fully available?

Reviewer #1: Yes

Reviewer #2: Yes

4. Is the manuscript presented in an intelligible fashion and written in standard English?

Reviewer #1: Yes

Reviewer #2: Yes

5. Review Comments to the Author

Reviewer #1: Thank you for your efforts with this research and manuscript. Your work addresses an important gap in techniques for in vitro rearing of Apis mellifera.

I have provided comments on two minor points below:

Line 55: I would suggest bullet pointing the three standardised pesticide risk assessments for honey bees to make the manuscript more easy to read.

Line 191: Was there any reason that you did not use an overall total diet of 400 μL to match the 2.5 fold difference between worker and drone weights?

Reviewer #2: This study successfully developed an in vitro rearing method for honey bee drones, from larvae to adulthood, and conducted a series of evaluations on this method. The novelty of the study is significant, particularly due to the multiple improvements made in the rearing process and the environmental conditions for honey bee males. However, the article could benefit from further refinement and optimization in certain details.

Figure legend is missing.

Authors might explain is that a possible approach for varroa culture in vitro?

Authors suggest provide a better explanation for why culture vertical or horizontal, any influence on the larvae development?

The authors could have included additional advantages and disadvantages of other methods for rearing male bees in the introduction section.

In the abstract and introduction, the authors used extensive language to describe the hazards of pesticides on honey bee worker bees and drones. However, the methodological design lacks a section dedicated to measuring the hazards of pesticides on individual honey bee worker bees and drones.

The article mentions transferring larvae from the hive to an in vitro rearing environment, but the details of the transfer process are insufficient. Key factors such as handling time, temperature conditions during transfer, and the potential impact of external pathogens or contaminants are not addressed. These factors could significantly influence larval survival and subsequent development. It is recommended that these details be added to the methods section.

What is the specific purpose of Figure S1? In lines 278-281, the authors compare the survival rates of grafted larvae after four and five days of queen caging. However, in lines 282-286, they reclassify this result as a comparison of survival rates for grafted larvae after four days of queen caging. Does this present a logical inconsistency or confusion?

I find it difficult to understand the necessity of including Honey bee workers in Figure 1. The authors present surprising results comparing the variations of drone pupae in different absorbent tissues and rearing orientations. However, the absence of results for worker bees and a lack of comparisons with drone data is quite confusing.

Lines 314-316, B and C are almost identical in description.

Each group has a different sample size. It is crucial to assess whether the experimental data meet the assumptions of normality. If one-way ANOVA is used, it should be specified whether the homogeneity of variance has been tested. If these assumptions are not met, it should be clarified whether non-parametric tests were employed. This information should be clearly included in the data analysis section.

6. PLOS authors have the option to publish the peer review history of their article (what does this mean?). If published, this will include your full peer review and any attached files.

Reviewer #1: No

Reviewer #2: **Yes: **Kai Wang

---

## [Author Response · Author response to Decision Letter 0]

1 Nov 2024

Thank you. The templates were consulted, and changes were made considering the journal requirements. 

2. We notice that your supplementary figures are uploaded with the file type 'Figure'. Please amend the file type to 'Supporting Information'. Please ensure that each Supporting Information file has a legend listed in the manuscript after the references list.

Thank you. All the supporting information files were renamed, and the legends are listed after the reference list.

Thank you. We appreciate your guideline link. The supporting information captions are available in the end of the manuscript.

Reviewer #1: 

R1.1: Thank you for your efforts with this research and manuscript. Your work addresses an important gap in techniques for in vitro rearing of Apis mellifera.

A1.1: Thank you. We appreciate your input; it will enhance our manuscript.

R1.2: Line 55: I would suggest bullet pointing the three standardized pesticide risk assessments for honey bees to make the manuscript more easy to read.

A1.2: Thank you. We have inserted the bullets in lines 59 to 64. 

R1.3: Line 191: Was there any reason that you did not use an overall total diet of 400 μL to match the 2.5 fold difference between worker and drone weights?

A1.3: Thank you for the great question. We didn’t have time to test for this in the current manuscript. In our upcoming study, we revised our approach, and we are still analyzing the data. Nonetheless, we believe that the 367 μl diet, which supported over 70% survival to adulthood, is valuable and worth publishing. However, as you pointed out, we could aim to match the recommended 2.5-fold increase by providing 400 μL of diet.

Reviewer #2: 

R2.1: This study successfully developed an in vitro rearing method for honey bee drones, from larvae to adulthood, and conducted a series of evaluations on this method. The novelty of the study is significant, particularly due to the multiple improvements made in the rearing process and the environmental conditions for honey bee males. However, the article could benefit from further refinement and optimization in certain details.

A2.1: Thank you. We sincerely appreciate your comment, and your input into our manuscript to improve it.

R2.2: Figure legend is missing.

A2.2: Thank you. We inserted the figure legend in figure 4 (lines 337 and 338). 

R2.3: Authors might explain is that a possible approach for varroa culture in vitro?

A2.3: Thank you for the great question. As we indicated in the introduction, our primary reason for developing an in vitro rearing protocol for honey bee drones is to facilitate pesticide exposure studies and pesticide risk assessments, which we plan to explore further in a future manuscript. However, we agree that, since Varroa mites preferentially parasitize drone brood, in vitro rearing of honey bee drones could also be a valuable approach for culturing Varroa mites, as you suggested. We look forward to exploring this potential application in the future.

R2.4: Authors suggest providing a better explanation for why culture vertical or horizontal, any influence on the larvae development?

A2.4: Thank you. We observed that when we transferred honey bee drone prepupae to the pupation plate that was placed horizontally, they tended to remain at the bottom of the wells, which we assume may reduce the space available for proper pupation. In contrast, positioning the plate vertically seemed to help the prepupae to extend length wise along the well, providing more space for their pupation. During the larval stage, we did not alter the plate orientation for several reasons: 1) The larvae have sufficient room to grow at the bottom of the well during this stage; 2) Changing the plate orientation could cause diet leakage from the wells; and 3) We handle the larval plate daily for feeding, which would infrequent alteration of the larval position. 

R2.5: The authors could have included additional advantages and disadvantages of other methods for rearing male bees in the introduction section.

A2.5: Thank you. We incorporated your suggestion from line 99 to line 105. 

R2.6: In the abstract and introduction, the authors used extensive language to describe the hazards of pesticides on honey bee worker bees and drones. However, the methodological design lacks a section dedicated to measuring the hazards of pesticides on individual honey bee worker bees and drones.

A2.6: Thank you for the insightful comment. While we introduced pesticide risk assessment in the introduction, the objective of this manuscript was not to measure the hazard of pesticides to honey bee drones. Rather, our aim was to develop a rearing method for honey bee drones that achieves over 70% survival, which in the future could be used in pesticide risk exposure studies. 

R2.7: The article mentions transferring larvae from the hive to an in vitro rearing environment, but the details of the transfer process are insufficient. Key factors such as handling time, temperature conditions during transfer, and the potential impact of external pathogens or contaminants are not addressed. These factors could significantly influence larval survival and subsequent development. It is recommended that these details be added to the methods section.

A2.7: Thank you. Your suggestions were incorporated from line 140 to line 145.

R2.8: What is the specific purpose of Figure S1? In lines 278-281, the authors compare the survival rates of grafted larvae after four and five days of queen caging. However, in lines 282-286, they reclassify this result as a comparison of survival rates for grafted larvae after four days of queen caging. Does this present a logical inconsistency or confusion?

A2.8: Thank you for the helpful suggestion to clarify our manuscript. The purpose of Figure S1 is to show that we tested drone survival when grafted four days after queen caging, as reported in the protocol for workers. In Figure S1, as indicated in the legend (lines 491–495), we compared the survival of drones grafted on day four in either horizontal or vertical orientations. However, Figure S1 does not compare drones grafted on day four with those grafted on day five. You are correct that we stated in lines 278–281 that drones grafted on day five had better survival than those grafted on day four; however, this comparison is not shown in Figure S1. To improve clarity, we have referenced the relevant figure in the text and added a sentence describing Figure S1.

R2.9: I find it difficult to understand the necessity of including Honey bee workers in Figure 1. The authors present surprising results comparing the variations of drone pupae in different absorbent tissues and rearing orientations. However, the absence of results for worker bees and a lack of comparisons with drone data is quite confusing.

A2. 9: Thank you. It is a good comment. Initially, we considered including the worker bee protocol in Figure 1 to help readers understand the differences between our protocol and worker bees' protocol. However, we agree that presenting the worker bee protocol in Figure 1 without showing corresponding results for workers could lead to confusion. Therefore, we have decided to remove the worker protocol from Figure 1.

R2.10: Lines 314-316, B and C are almost identical in description.

A2.10: Thank you. We changed it on lines 323 and 324. 

R2. 11: Each group has a different sample size. It is crucial to assess whether the experimental data meet the assumptions of normality. If one-way ANOVA is used, it should be specified whether the homogeneity of variance has been tested. If these assumptions are not met, it should be clarified whether non-parametric tests were employed. This information should be clearly included in the data analysis section.

A2.11: Thank you. We assessed normality using the Shapiro-Wilk test, as described on lines 272 and 273. Since we are comparing only two groups (PVW group reared in vitro and control drones reared in field colonies) for body weight, testes weight, and abdominal area, we did not use one-way ANOVA; instead, we performed t-tests because the data was normally distributed for all groups as indicated in lines 276 to 279. Variance was evaluated using Levene's test. For mean prepupal body weight and mean adult testes weight, the assumption of equal variance was not met (in lines 279 to 284), we applied Welch's t-test. The mean of adult body weight and mean of adult abdominal area the equal variance assumption was met (in lines 284 to 288), we used the independent t-test.

---

## [Decision Letter · Decision Letter 1]

19 Nov 2024

Re: From larva to adult: *In vitro* rearing protocol for honey bee (*Apis mellifera*) drones

PONE-D-24-39196R1

Dear Dr. Bezerra da Silva,

We’re pleased to inform you that your manuscript has been judged scientifically suitable for publication and will be formally accepted for publication once it meets all outstanding technical requirements.

An invoice will be generated when your article is formally accepted. Please note, if your institution has a publishing partnership with PLOS and your article meets the relevant criteria, all or part of your publication costs will be covered. Please make sure your user information is up-to-date by logging into Editorial Manager at Editorial Manager^®^ and clicking the ‘Update My Information' link at the top of the page. If you have any questions relating to publication charges, please contact our Author Billing department directly at authorbilling@plos.org.

Kind regards,

Yahya Al Naggar

Academic Editor

PLOS ONE

Additional Editor Comments (optional):

Reviewers' comments:

Reviewer's Responses to Questions

**Comments to the Author**

1. If the authors have adequately addressed your comments raised in a previous round of review and you feel that this manuscript is now acceptable for publication, you may indicate that here to bypass the “Comments to the Author” section, enter your conflict of interest statement in the “Confidential to Editor” section, and submit your "Accept" recommendation.

Reviewer #2: (No Response)

2. Is the manuscript technically sound, and do the data support the conclusions?

Reviewer #2: (No Response)

3. Has the statistical analysis been performed appropriately and rigorously? 

Reviewer #2: (No Response)

4. Have the authors made all data underlying the findings in their manuscript fully available?

Reviewer #2: (No Response)

5. Is the manuscript presented in an intelligible fashion and written in standard English?

Reviewer #2: (No Response)

6. Review Comments to the Author

Reviewer #2: (No Response)

7. PLOS authors have the option to publish the peer review history of their article (what does this mean?). If published, this will include your full peer review and any attached files.

Reviewer #2: **Yes: **Kai Wang

---

## [Editor Report · Acceptance letter]

22 Nov 2024

PONE-D-24-39196R1 

PLOS ONE

Dear Dr. Bezerra da Silva, 

I'm pleased to inform you that your manuscript has been deemed suitable for publication in PLOS ONE. Congratulations! Your manuscript is now being handed over to our production team.

Kind regards, 

on behalf of

Dr. Yahya Ahmed Shaban Al Naggar 

Academic Editor

PLOS ONE